# Population heterogeneity in associations between hormonal contraception and antidepressant use in Sweden: a prospective cohort study applying intersectional multilevel analysis of individual heterogeneity and discriminatory accuracy (MAIHDA)

Sofia Zettermark [1], Kani Khalaf,[1] Raquel Perez-Vicente,[1] George Leckie,[1,2] Diana Mulinari,[3] Juan Merlo [1,4]

For numbered affiliations see end of article.

**Correspondence to**
Sofia Zettermark;
sofia.zettermark@med.lu.se

## ABSTRACT

**Objectives** From a reproductive justice framework, we aimed to investigate how a possible association between hormonal contraceptive (HC) and antidepressants use (as a proxy for depression) is distributed across intersectional strata in the population. We aimed to visualise how intersecting power dynamics may operate in combination with HC use to increase or decrease subsequent use of antidepressants. Our main hypothesis was that the previously observed association between HC and antidepressants use would vary between strata, being more pronounced in more oppressed intersectional contexts. For this purpose, we applied an intersectional multilevel analysis of individual heterogeneity and discriminatory accuracy approach.

**Design** Observational prospective cohort study using record linkage of national Swedish registers.

**Setting** The population of Sweden.

**Participants** All 915 954 women aged 12–30 residing in Sweden 2010, without a recent pregnancy and alive during the individual 1-year follow-up.

**Primary outcome measure** Use of any antidepressant, meaning being dispensed at least one antidepressant (ATC: N06A) during follow-up.

**Results** Previously mentally healthy HC users had an OR of 1.79 for use of antidepressants compared with non-users, whereas this number was 1.28 for women with previous mental health issues. The highest antidepressant use were uniformly found in strata with previous mental health issues, with highest usage in women aged 24–30 with no immigrant background, low income and HC use (51.4%). The largest difference in antidepressant use between HC users and non-users was found in teenagers, and in adult women of immigrant background with low income. Of the total individual variance in the latent propensity of using antidepressant 9.01% (healthy) and 8.16% (with previous mental health issues) was found at the intersectional stratum level.

---

### Strengths and limitations of this study

► Entire Swedish population of women aged 12–30 included.

► Pharmacy dispensing automatically linked to individual personal identification number in Sweden through the Swedish Prescribed Drug Register and thus very reliable.

► Intersectional multilevel analysis of individual heterogeneity and discriminatory accuracy is a fruitful way of epidemiologically investigating heterogeneity within a population while considering individual conditions determined by societal power dimensions such as class, gender and race.

► Antidepressant dispensing is not a perfect proxy for depression.

► Registers cannot measure actual use of any medication.

---

**Conclusions** Our study suggests teenagers and women with immigrant background and low income could be more sensitive to mood effects of HC, a heterogeneity important to consider moving forward.

## INTRODUCTION

In recent years, attention in the medical community has increasingly been drawn towards depression and other adverse effects on mood related to use of hormonal contraception (HC).[1 2] Discontinuation rates are high, with mood disturbances or depression being one of the most common complaints.[3–5] Two large epidemiological studies, one in Denmark and the other performed in Sweden, have recently shown a higher risk of antidepressants and psychotropic

drugs use in adolescent users of HC.[6 7] Randomised controlled trials are rare, but suggest a negative influence of HC on well-being and sexual function,[8 9] as well as evidence of HC modulating brain activity with subsequent mood alterations in some women.[10 11] Even though oestrogen and progesterone are known to affect mood,[12] the growing body of evidence in this field is contradictory, with recent reviews concluding that both protective and negative effects of HC on mood exist and more research is needed.[13–16] Despite this uncertainty, many scholars agree that certain subgroups of women seem more vulnerable to psychological side effects of HC than others, particularly teenagers and women with previous mental health issues.[10 13 17–20] A call for further investigation into these vulnerable subgroups has been made.[14]

A fruitful way of epidemiologically investigating heterogeneity within a population while considering individual conditions determined by societal power dimensions such as class, gender and race has been developed through intersectional theory in recent years.[21–26] Intersectionality theory was first articulated by Black feminist scholars as a way of understanding how an individual inhabits and is formed by more than one social relation such as gender, 'race' or class, and how these classification systems interconnect to create specific contexts of oppression or privilege.[27 28] These categorisations should not be seen as individual 'risky' identities, but as the social, political and economic contextual conditions that outline our lives through structural inequalities.[29] Reproductive justice is a theoretical framework that builds on intersectionality and centres diverse groups of unprivileged women's reproductive experiences to recognise that societal context and differing resources available shape reproductive health.[30] Applying a reproductive justice framework, it becomes clear that we need to take notice of disparate sociocultural contexts and interlocking power dimensions to understand different patterns of usage as well as possible diverse responses to HC.[31 32]

To operationalise an intersectional mapping of heterogeneity in use of antidepressants in relation to HC on a population level, we used a multilevel analysis of individual heterogeneity and discriminatory accuracy (MAIHDA).[21–23 33 34] We created intersectional strata based on previous literature showing that age, socioeconomic position and previous mental illness are relevant intersecting dimensions in understanding the relation between HC and depression.[17 20 35 36]

We conceptualise the intersectional strata as social contexts rather than static individual traits, thereby visualising how intersecting power dynamics can act in combination with HC to predispose for depressive mood. Our main hypothesis was that the previously observed association between HC and use of antidepressants would vary between strata and that this association would be more pronounced in more oppressed intersectional contexts. We investigate this hypothesis on the whole population of women susceptible to HC use in Sweden.

## METHOD

### Databases and study population

After allowance from the Swedish Ethical Authority and the data safety committees from Statistics Sweden and the Swedish National Board of Health and Welfare, we obtained a database created by record linkage of several nationwide registers administered by Statistics Sweden (the Swedish Population Register and the Longitudinal Integration Database for Health Insurance and Labour Market Studies, LISA) and the Swedish National Board of Health and Welfare (National Patient Register, the Swedish Prescribed Drug Register (SPDR) and the Cause of Death Register). The Swedish authorities linked the registries using a unique personal identification number, but the database was anonymised before delivering it to us.

We defined an initial cohort containing all 1 064 171 women aged 12–30 years residing in Sweden 1 January 2010 and obtained individual level data on medication use from SPDR, which contain all dispensed drug prescriptions at Swedish pharmacies since 2006.

Every woman was assigned an individual baseline date, defined by the first dispensed prescription of an HC drug between 1 January 2010 and 31 December 2014 after 12 years of age, and was then followed for 1 year after her individual baseline date. A woman obtaining her first prescription on 1 September 2013 was therefore followed to 1 September 2014. For non-users of HC, the baseline date could not be based on a HC-prescription and was therefore assigned, to 1 July 2012 for all adults, but later for some of the younger girls turning 12 during our period of investigation. This means all non-users had been true non-users for at least 1.5 years before their follow-up started (1 January 2010 to 1 July 2012) but also continued to be non-users all the way to 31 December 2014. From the individual baseline date, the women were followed for 1 year to find out if a prescription of an antidepressant was dispensed. Data were also collected on psychiatric disorders and psychotropic drug use in the past 3 years (see Assessment of variables). After excluding women with incomplete follow-up time due to death, emigration, missing information on country of birth, and pregnancies 1 year before and after the baseline as well as, the final database consisted of 915 952 women. This database was divided into two cohorts according to the presence or absence of previous mental health issues, see figure 1.

### Assessment of variables

Users of HC were defined as any women who, according the SPDR, filled a prescription of HC (Anatomical Therapeutical Chemical (ATC) classification system codes G02B, G03AA-C) between 1 January 2010 and 31 December 2014, while non-users did not have a prescription filled during the same period. Emergency contraception (G03AD) that are mainly bought over the counter in Sweden was excluded. The majority of HC prescriptions are acquired via midwifes in Sweden (86.0% in our original cohort), whom can only prescribe HC for

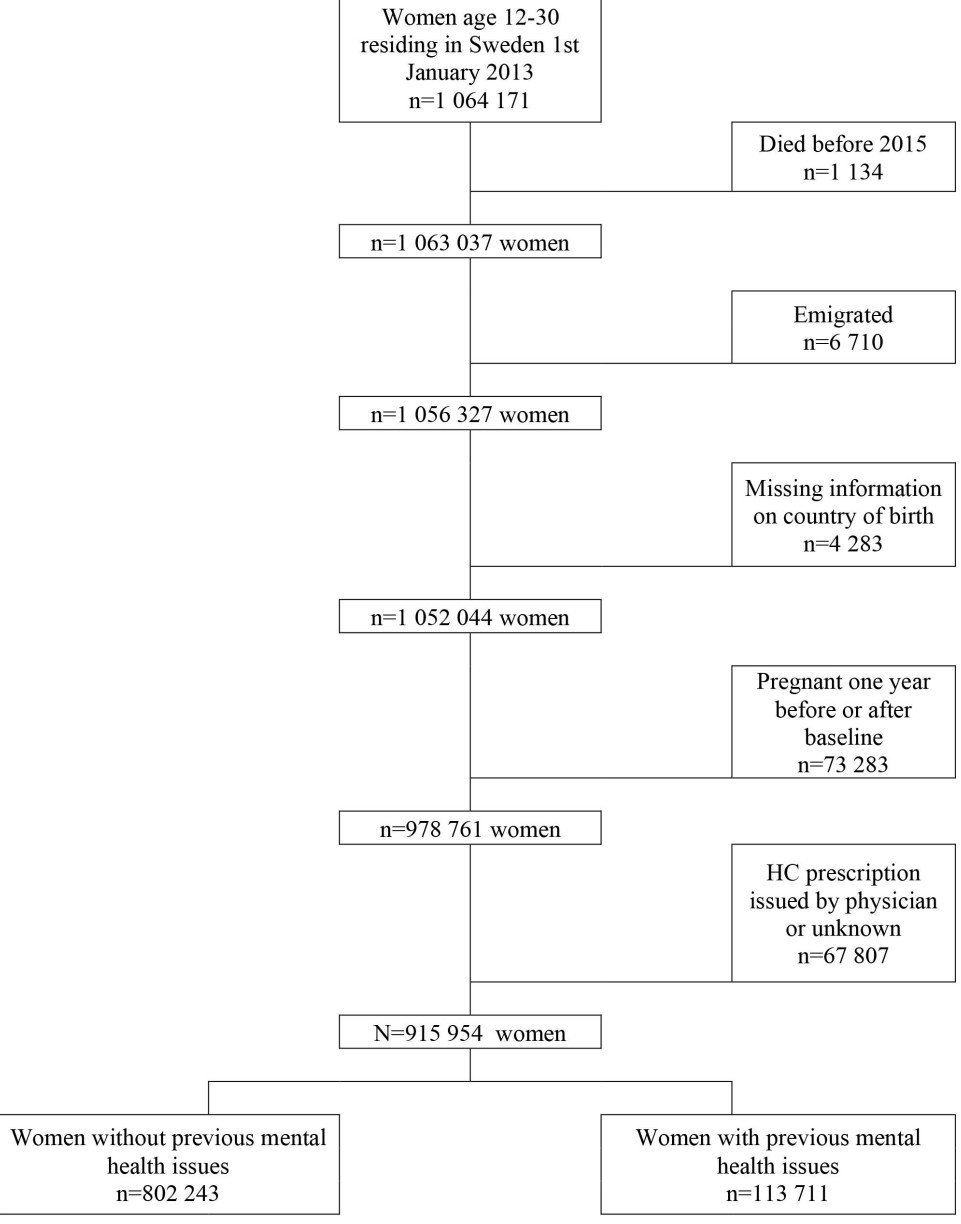

**Figure 1** Selection of the study population. HC, hormonal contraception.

contraceptive purposes. Physicians, most often gynaecologists, can also can prescribe HC for other purposes such as in response to bleeding disturbances or endometriosis. Since these indications could confound our results, we excluded women with physician-issued prescriptions, see figure 1. HC prescriptions can be dispensed by pharmacies annually or every 3 months.

Antidepressant use, the outcome of our study, was defined, according to the SPDR, as being dispensed at least one prescription of antidepressants (ATC: N06A) during the individual 1-year follow-up.

Previous mental health issues were defined as having any psychiatric disorder diagnosed at a hospital (ICD: F00-F99) or a dispensed prescription of a psychotropic drug (ATC: N05A, N05B, N06A) in the past 3 years.

Pregnancies 1 year previous to baseline and during follow-up were identified according to the 2019 version

of the Nordic Diagnosis-Related Group classification (NordDRG), Major Diagnostic Categories codes M14 for pregnancy, delivery and postpartum care.[37]

We used family level data on income as of 31 December 2010 from Statistics Sweden's LISA. Individualised disposable family income was calculated by dividing the total disposable income of the family by the number of family members, taking into account the different consumption weights of adults and children determined by Statistics Sweden. Thereafter, we created three categories (ie, low, medium and high) of income using tertile cut-offs based on the total Swedish population aged 18–80 years. We considered the high-income category as the reference in the comparisons.

We defined immigrant status at the family level as no family member >18 years of age born in Sweden, since understanding of and access to institutions such as

healthcare differ depending on social position such as it is constructed by the power dimensions of race/immigration, as well as the experience of xenophobia. This variable should therefore be considered as an effort to capture a social position affecting possibilities and life trajectories rather than an essentialist view of otherness. We categorised age at the individual baseline into the following groups: 12–17, 18–23 and 24–30 years to capture age specific conditions of adolescents, young adults and adult women.

### Intersectional strata

Within each cohort stratified by previous mental health issues, we generated 36 intersectional strata by combining three categories of age, three categories of income, two categories of immigrant background and two categories of HC use. Mental health issues can be considered as a valid category of intersectional investigation in a society that considers an able body and mind vital, in other words relating to the power dimension of able-bodiedness.[38 39] Mental health issues were also included in the analysis since they are a strong determinant of antidepressant use that needs to be addressed. We could consider that over and above individual characteristics, mental illness-related stigma may condition inequities in healthcare.[40] As with gender or income, able-bodiedness concerning mental health can therefore be conceptualised as a contextual dimension when defining intersectional strata.

### Statistical analysis

We performed an intersectional MAIHDA with individual women at the first level of analysis and the 36 intersectional strata at the second level, stratified by previous mental health issues (see online supplemental material 1–4). The use of antidepressants in the population was thus analysed through two successive multilevel logistic regression models distinguishing between measures of association and measures of variance and discriminatory accuracy.

### Model 1

The first model included only an intercept and a random effect for the intersectional strata with no covariates. In this model 1, we first (i) performed a simple analysis of components of variance and calculated the variance partition coefficient (VPC). That is, the share (expressed as a percentage) of the total individual variance in the latent propensity of antidepressant use, that is, at the intersectional strata level. In this simple model, the VPC correspond with the intraclass correlation coefficient which informs on the clustering of antidepressant use within intersectional strata. The VPC values extend from 0% to 100%. Second, (ii) we calculate the stratum-specific absolute usage of antidepressants and their 95% credible intervals (CI) by transformation of the information from the logistic regression to the probability scale. We used this information to map the user heterogeneity across the intersectional strata. Then, (iii) using these stratum-specific predictions, we calculated the Area Under the receiver operator characteristics Curve (AUC). The AUC informs on the accuracy of the intersectional strata information for discriminating those women who used antidepressants from those who did not. The AUC values extend from 0.5 to 1, where 0.5 represents absence of accuracy and 1 represents total accuracy. Both the VPC and the AUC in model 1 can be interpreted as measures of discriminatory accuracy,[41] and inform on the magnitude of the general intersectional effects. The higher the VPC and AUC values, the higher the influence of the intersectional context on individual use of antidepressants. Finally, (iv) we calculated the difference in antidepressant use and 95% CI between similar pairs of strata differing only on the use of HC. This represents the stratum specific association between HC and antidepressant use.

### Model 2 or fixed main effects model

This model includes the fixed, main effects of all the intersectional dimensions (ie, age, income, immigrant background and HC use) used to define the intersectional strata. In model 2, we quantified, (i) the association between the intersectional dimensions and use of antidepressants as expressed by OR and 95% CI. We also to calculate (ii) the proportional change in the variance (PCV). The PCV measures the overall proportion of strata variance of model 1 explained by the specific intersectional dimensions. Since model 2 contains all the variables used to construct the intersectional strata as main effects, it should explain all the strata variance (ie, PCV=100%). If this is not the case, the remaining between strata variance would be due to the existence of multiplicative interaction of effects between the intersectional dimensions defining the strata.[22 42]

The AUCs of the models 1 and 2 are expected to be the same because model 2 only decomposes the stratum-specific predicted probabilities obtained in model 1 into fixed and random-effect components and their sum equals the prediction obtained only by random effects in model 1.

We ran the models using MLwiN V.3.00 by calling it from within Stata V.14.1 using the *runmlwin* command.[43] The estimations were performed using Markov chain Monte Carlo (MCMC) methods. All point estimations and their 95% CIs were based on the parameter and random-effect chains obtained from the MCMC estimation. See elsewhere for further information on the statistical MAIHDA analysis including Stata commands,[33 42] and discussion on the theory and methodological approach.[22 44]

### Patient and public involvement statement

The research was developed with a grassroot perspective in mind, whereby women's experiences of use of HC inspired and informed the choice of research area and research questions. The anonymised data and scope of the study, including around 1 million women, prohibited direct patent involvement.

**Table 1** Characteristics of the 915 954 women aged 12–30 years by previous mental health issues and use of hormonal contraceptives

| | Previous mental health issues | | | |
|---|---|---|---|---|
| | Yes 12.4 (113 711) Use of HC | | No 87.6 (802 243) Use of HC | |
| | Yes 42.5 (48 302) | No 57.5 (65 409) | Yes 42.0 (337 297) | No 58.0 (464 946) |
| Antidepressant during follow-up | 41.2 (19 886) | 39.8 (26 013) | 2.7 (9215) | 1.9 (8699) |
| Age | | | | |
| 12–17 years | 14.2 (6838) | 19.4 (12 698) | 16.7 (56 343) | 42.1 (195 937) |
| 18–23 years | 48.3 (23 347) | 31.2 (20 381) | 50.1 (168 968) | 23.3 (108 939) |
| 24–30 years | 37.5 (18 117) | 49.4 (32 330) | 33.2 (11 986) | 34.6 (160 616) |
| Income level | | | | |
| Low income | 40.4 (19 513) | 45.6 (29 803) | 31.8 (107 119) | 33.1 (154 098) |
| Medium income | 27.1 (13 078) | 27.5 (17 954) | 25.4 (85 620) | 29.5 (137 098) |
| High income | 32.5 (15 711) | 27.0 (17 652) | 42.9 (144 558) | 37.4 (173 750) |
| Immigrant background | | | | |
| None | 94.6 (45 674) | 89.1 (58 264) | 94.2 (317 716) | 82.6 (383 878) |
| Yes | 5.4 (2628) | 10.9 (7145) | 5.8 (19 581) | 17.4 (81 068) |

Values are percentages (number of women in parenthesis).
HC, hormonal contraception.

## RESULTS

### Characteristics of the population

The selection of the study population is shown in figure 1. Out of the 915 952 women, 12.4% (n=113 711) had previous mental health issues. Mean age was somewhat older for women with previous mental health issues (22.5 years; SD 4.8) than for those without such concerns (20.8 years; SD 5.3). Online supplemental material 5 shows pooled statistics for usage of previous mental health issues and HC use, while online supplemental material 6 displays a frequency table over all included HC. Table 1 displays the baseline characteristics of the population by previous mental health issues and use of HCs.

The share of HC users was very similar in healthy women and those with previous mental health issues, 42.0% and 42.5%, respectively. Antidepressants were dispensed to 2.7% of HC users compared with 1.9% of non-users among healthy women during follow-up. For women with previous mental health issues, 41.2% of HC users and 39.8% of non-users dispensed an antidepressant prescription. The income levels were generally higher among women without mental health issues, and HC users were somewhat more affluent in both cohorts.

### Results from the MAIHDA

Table 2 shows the results from the MAIHDA distinguishing between measures of association and measures of variance and discriminatory accuracy.

Model 1 indicates that 8.45% (without mental health issues) and 8.18% (with previous mental health issues) of the total individual variance in the latent propensity of using antidepressant is at the intersectional strata level.

These VPCs correspond with AUC values of 0.62 and 0.64, respectively. Both measures suggest the existence of a moderate intersectional effect. The PCV was high in both groups, but especially so in the group with previous mental health issues, meaning the intersectional dimensions or main effects explain more of the interstrata variance for these women. Model 2 shows that HC was associated with increased usage of antidepressants after adjustment for all other intersectional dimensions. This result was seen within both cohorts, but more strongly so in women without previous mental health issues (OR 1.62 compared with 1.19). Finally, the VPC in model 2 was very small (3.02% and 0.49%, respectively) but did not vanish. This finding means that while the intersectional strata effect was mainly due the additive effect of variables defining the strata, a small component due to interaction of effects could also be detected.

### Heterogeneity concerning antidepressant use in our cohort

Women with previous mental health issues had a much higher usage of antidepressants than women without such issues, but the association with HC use nonetheless varied across the other intersectional dimensions. Table 3 shows the stratum-specific incidence rates for antidepressant use and 95% CI obtained in model 1.

The highest use of antidepressants were observed in non-immigrant women, aged 24–30, with previous mental health issues, using HC and with low income (50.1%). The lowest usage were found in teenagers without previous mental health issues and no HC use, especially in the strata of immigrant girls from low (0.50%) and middle-income (0.60%) households.

**Table 2** Results from the multilevel analysis of individual heterogeneity and discriminatory accuracy (MAIHDA) distinguishing between measures of association (ORs) and measures of variance and discriminatory accuracy.

| | Without mental health issues | | With mental health issues | |
| --- | --- | --- | --- | --- |
| | Model 1 | Model 2 | Model 1 | Model 2 |
| Measures of association | | | | |
| Age | | | | |
| 12–17 years | | Reference | | Reference |
| 18–23 years | | 1.78 (1.36–2.42) | | 1.57 (1.38–1.76) |
| 24–30 years | | 2.09 (1.65–2.70) | | 2.66 (2.36–3.00) |
| Income | | | | |
| High income | | Reference | | Reference |
| Medium income | | 1.05 (0.78–1.37) | | 0.87 (0.77–0.98) |
| Low income | | 1.10 (0.81–1.41) | | 0.87 (0.77–0.98) |
| Immigrant background | | | | |
| None | | Reference | | Reference |
| Yes | | 0.63 (0.49–0.79) | | 0.55 (0.49–0.61) |
| Hormonal contraception | | | | |
| No | | Reference | | Reference |
| Yes | | 1.62 (1.34–2.06) | | 1.19 (1.08–1.31) |
| Measures of variance and discriminatory accuracy* | | | | |
| Variance | 0.30 (0.18–0.50) | 0.10 (0.06–0.18) | 0.29 (0.18–0.49) | 0.02 (0.01–0.03) |
| VPC | 8.45% | 3.02% | 8.18% | 0.49% |
| PCV | | 66.29% | | 94.48% |
| AUC | 0.62 (0.62–0.62) | 0.62 (0.62–0.62) | 0.64 (0.64–0.64) | 0.64 (0.64–0.64) |

The analyses are stratified by the existence of previous mental issues.
Values are point estimations (with 95% credible intervals) or percentages where indicated.
*Between-strata variance.
AUC, area under the curve; PCV, proportional change of the variance; VPC, variance partition coefficient.

### Heterogeneity concerning the association between hormonal contraceptive and antidepressant use

Overall, the propensity to use antidepressants was consistently higher in HC users compared with non-users in younger women between 12 and 17 years of age, both without previous mental health issues (0.7–2.4 percentage points), and with a mental health history (5.7–7.8 percentage points) with the magnitude being higher in the latter group. However, the 95% CIs were broad since the number of individuals was relatively small in these latter strata. Table 3 gives detailed information on these associations. In adolescents, the tendency was that an immigrant background lowered the use of antidepressants, while the opposite was true for adult women, where a positive association between HC use and later antidepressant use was mainly found in women with low income and immigrant background, again with higher magnitudes in women with previous mental health issues. The association between HC and antidepressant use was smaller in adult women native to Sweden no matter their income, and completely disappeared in adult women with high income regardless of immigrant background.

### DISCUSSION

The main hypothesis of our study was that the previously observed association between HC and antidepressant use, mainly seen in adolescent girls,[6–9 17 45] would be modified by the intersectional context of the women, being more pronounced in more oppressed intersectional contexts. We confirmed that subsequent use of antidepressants after an HC prescription compared with non-users of HC within the same intersectional context was heterogeneous across intersectional strata pairs. As hypothesised, the difference in propensity to use antidepressants was more pronounced in more oppressed intersectional contexts like those composed by immigrant, low-income women with previous mental issues. That is, the use of antidepressants and to some extent the difference in use between HC users and non-users varied mainly depending on previous mental health issues, but the HC-antidepressant association was considerably modified across pair of strata with other characteristics equal but where HC use and non-use differed, in both cohorts. Aside from adolescent girls, low-income and middle-income adult women with immigrant background had a more pronounced

**Table 3** Distribution of antidepressant use between different intersectional strata, and difference in usage between user and non-users of hormonal contraceptives but otherwise sharing the same intersectional stratum.

| Previous mental health issues | Age (years) | Income level | Immigrant background | Number of women | Use of hormonal contraceptives (%) | | |
|---|---|---|---|---|---|---|---|
| | | | | | Yes | No | Yes–no difference |
| No | 12–17 | Low | No | 28 182 | 3.7 | 1.3 | **2.4 (1.9, 2.8)** |
| | | | Yes | 7643 | 1.2 | 0.5 | **0.7 (0.1, 1.5)** |
| | | Middle | No | 75 836 | 3.0 | 1.0 | **2.0 (1.8, 2.3)** |
| | | | Yes | 10 110 | 1.8 | 0.6 | **1.2 (0.5, 2.1)** |
| | | High | No | 125 903 | 2.0 | 0.9 | **1.1 (0.9, 1.2)** |
| | | | Yes | 4606 | 2.5 | 0.8 | **1.6 (0.6, 2.8)** |
| | 18–23 | Low | No | 44 723 | 3.5 | 3.0 | **0.5 (0.2, 0.9)** |
| | | | Yes | 11 174 | 2.3 | 1.2 | **1.1 (0.5, 1.7)** |
| | | Middle | No | 72 018 | 2.8 | 2.8 | 0.1 (−0.2, 0.3) |
| | | | Yes | 8776 | 2.3 | 1.2 | **1.1 (0.5, 1.8)** |
| | | High | No | 136 284 | 2.3 | 2.3 | 0 (−0.2, 0.1) |
| | | | Yes | 4386 | 2.0 | 1.8 | **0.2 (−0.6, 0.9)** |
| | 24–30 | Low | No | 130 127 | 3.1 | 3.2 | −0.1 (−0.3, 0.1) |
| | | | Yes | 39 368 | 2.7 | 1.4 | **1.3 (0.9, 1.7)** |
| | | Middle | No | 45 013 | 3.6 | 3.0 | **0.5 (0.2, 0.9)** |
| | | | Yes | 10 965 | 2.7 | 2.4 | 0.4 (−0.3, 1.1) |
| | | High | No | 43 508 | 2.4 | 2.6 | −0.2 (−0.5, 0.1) |
| | | | Yes | 3621 | 1.9 | 2.3 | −0.3 (−1.3, 0.7) |
| Yes | 12–17 | Low | No | 3402 | 30.5 | 22.7 | **7.8 (4.7, 10.8)** |
| | | | Yes | 434 | 20.8 | 13.7 | 7.1 (−0.3, 15.1) |
| | | Middle | No | 6854 | 31.2 | 23.4 | **7.8 (5.6, 10.1)** |
| | | | Yes | 569 | 19.9 | 14.2 | 5.7 (−1.2, 13.1) |
| | | High | No | 7906 | 34.2 | 28.1 | **6.1 (3.9, 8.3)** |
| | | | Yes | 371 | 30.4 | 19.8 | **10.6 (1.4, 19.9)** |
| | 18–23 | Low | No | 10 937 | 39.2 | 37.8 | 1.4 (−0.4, 3.2) |
| | | | Yes | 1127 | 28.5 | 19.7 | **8.8 (3.4, 14.4)** |
| | | Middle | No | 12 915 | 37.8 | 36.3 | 1.5 (−0.2, 3.1) |
| | | | Yes | 844 | 27.4 | 19.7 | **7.7 (1.9, 13.7)** |
| | | High | No | 17 276 | 38.3 | 39.8 | −1.5 (−3, 0) |
| | | | Yes | 629 | 28.1 | 25.4 | 2.8 (−4, 9.4) |
| | 24–30 | Low | No | 29 333 | 50.1 | 49.9 | 0.2 (−1, 1.4) |
| | | | Yes | 4083 | 37.3 | 32.4 | **4.9 (1.5, 8.4)** |
| | | Middle | No | 8629 | 49.7 | 50.8 | −1.1 (−3.4, 1.1) |
| | | | Yes | 1221 | 33.5 | 37.1 | −3.6 (−10, 2.6) |
| | | High | No | 6686 | 48.5 | 48.9 | −0.4 (−2.9, 2) |
| | | | Yes | 495 | 43.7 | 37.5 | 6.3 (−3.2, 15.8) |

The values are calculated from the multilevel analysis of individual heterogeneity and discriminatory accuracy (MAIHDA).
Numbers are percentages.
Bold values indicate a statistically significant difference.

difference in propensity for using antidepressants, while adult women without immigrant background had both the lowest antidepressant use and a low grade of modification by HC use.

Independently of previous mental health issues, the propensity for using antidepressants was consistently higher for HC users than for non-users in teenagers aged 12–17, a result aligned with previous studies that has

found a heterogeneous response with regard to both age and other factors.[6 7 17 18 20 45–47] As discussed in a previous paper, this higher risk for adolescents could be due to a *selective discontinuation bias,*[7] a development of the *healthy worker survivor effect,* describing how bias is introduced through a continuous selection where those staying in the workforce are healthier than those who leave.[48] Women who experience a negative influence of HC on psychological health might discontinue treatment in early ages, while those without symptoms continued on HC into adulthood, creating this age-dependent *selective discontinuation bias.* This could explain why the observed association between HC and adverse mental health outcomes are stronger in adolescents. Most Swedish women do however continue their HC treatment with the same method.[49] A previous study found that new users of HC has a higher risk of obtaining antidepressants within the first 6 months of HC use than continuous users.[6] To address this possible bias, we ran two different sensitivity analyses differentiating between women who filed a first prescription of an HC for the first time during the study period (26.2% of HC users) and those that had a repeat prescription. In our cohort, the association between HC use and subsequent antidepressant use was very similar in new and continuous users, but slightly higher among new users, as expected (OR 1.52 and 1.45, respectively, with overlapping 95% CIs). We then excluded all women with HC use any time during 5 years before baseline, thus including using only new users of HC during baseline and never-users as reference group (n=532 543) and reran the analysis. The association between HC use and subsequent antidepressant use became somewhat stronger in women without mental health issues (OR 1.86) and the VPC also increased. The pattern of antidepressant use in the intersectional strata stayed the same, but the CIs increased since the number of women included was smaller, see online supplemental material 7.

As expected, among adult women the overall propensity for using antidepressants was higher, as it is known that antidepressant use increases by age,[50 51] and the difference between HC users and non-users was smaller. Women native to Sweden had a higher propensity for using antidepressants, but this was moderated by HC exposure to a lower extent than for immigrant women. In adult women native to Sweden, HC use gave no increase of antidepressant use among those with high income. The lower utilisation of antidepressants does not necessarily mean that immigrant women are healthier, since earlier studies have found immigrants utilise healthcare to a lesser extent, even though the need is pronounced, with reasons including discrimination.[52 53] A recent study found that adjusement for healthcare access eliminated the association between HC initiation and subsequent antidepressant use in a US population.[54] Although the healthcare system is different in Sweden and visits to midwifes for contraceptive purposes free, we conducted a sensitivity analysis including only women who had accessed healthcare within the last 3 years to adress this.

Using only care-accessors as the reference group did not change our results in any substansive way, see online supplemental material 8.

## Intersectional considerations

The big difference in antidepressant consumption depending on HC use for lower income immigrant women could be interpreted as the intersectional contexts embodied by these women are more susceptible to the potential detrimental effect of HC on mood. The interrelating negative consequences of low income as a proxy for class or social position, gender and xenophobia may accumulate over the life course and lead to a higher vulnerability to exposures that predispose for antidepressant use later in life,[55–57] whereas this diverse vulnerability to HC exposure might not be visible in teenagers. Social experiences can vary depending on, for example, social position, which in turn impact psychological development, mood and cognition, thus influencing health.[58 59] In understanding how HC can impact women's mental health differently, both possible individual biological predispositions and social settings need to be investigated, since the emotional response to HC is influenced by context.[32] In other words, the interlocking power axes that create oppression could predispose women already under structural burdens for adverse mental health reactions when using HC. The fact that adult women native to Sweden were almost unaffected by HC use, could strengthen this suggestion. Without the intersectional strata, this disparity would not have been so easily identified and visualised.

Focusing on women whose lives are affected by several interlocking power dimensions such as low social position and xenophobia is fundamental to achieving reproductive justice.[30] Nonetheless, our intersectional strata should not be considered static categories of inherently 'risky identities' but must be interpreted as context-specific vulnerabilities of women within certain interlocking positions, constituted in relation to power dynamics created by unequal schemes such as the economic system.[25 29] It is likely that in other contexts, other groups could be more vulnerable. It is also important to remember that the purpose of HC most commonly is protection against unwanted pregnancy, a situation that if it arises in itself can have negative mental health effects. In identifying the underlying power systems creating these intersectional categories and acknowledging their constant movement and changing dynamics on a societal level, it furthermore becomes possible to address these inequalities through social change.

In this study, we have combined a classical epidemiological approach of exposure to HC and an intersectional MAHIDA to create a novel understanding of how intersecting power dynamics could create particular vulnerabilities to this specific exposure. Because of our study design, where women are followed for 1 year after a dispensed prescription of HC, it is more theoretically coherent to view use of HC as an exposure rather than

a component of the intersectional strata. However, it is possible to within our approach view HC use as a socio-contextual factor that captures certain living conditions (eg, more likely to be sexually active or in a heterosexual relationship), which somewhat changes the interpretation of the results. This epistemological tension is not necessarily a limitation, but could enrich the dialogue in social epidemiology on whether it is possible to separate contextual factors from 'pure' exposure.[60–62]

## Limitations

The findings from this study must be interpreted in the context of its limitations. The SPDR has highly reliable data on dispensed prescriptions but cannot measure the actual use of dispensed medications. Whether the women was exposed to HC treatment during her entire follow-up is thus not possible to determine with our method, although previous Swedish data suggest continuation rates for any HC after 6 months are almost 90%.[47] Our methodology does furthermore not allow for differentiation between new users and continuous users of HC. Previous studies has shown an increased risk for depression in new users,[6] which could mean we underestimate the associations when also including continuous users. Nevertheless, a sensitivity analysis (see online supplemental material 7) showed that the pattern of antidepressant use and heterogeneity between groups that the MAIHDA shows remain the same when including only new users. Combining MAIHDA with a survival analysis would possibly address this issue better and could be considered in the future. Use of antidepressants can be considered a proxy for depression, but antidepressants are also prescribed for other reasons than depression, including generalised anxiety disorder, obsessive–compulsive disorder and panic disorder.[63] Therefore, it is not a perfect proxy of depression but may be a more general indication of impaired mental health.[64] However, out of all women with potentially unfavourable mental health effects from HC, only a subset would have symptoms severe enough to get an antidepressant prescription, leading instead to many missed cases. Since the outcome is rather common, the risk of underestimation is further enhanced and the true risk of adverse mental health effects could be higher.

As in any observational study, ours only allows for measurements of associations and cannot determine causation. Furthermore, apparently strong average associations do not necessarily convey a high discriminatory accuracy (see elsewhere for a short review and discussion).[65] Nevertheless, since our analysis yielded a moderate accuracy (ie, AUC=0.6), the intersectional strata do matter for the propensity to use antidepressants. A consideration in every quantitative intersectional study is the basis for creating intersectional categories, since comprehensive information on background and lived experiences are lacking and the categories are created based on available but crude proxies such as income level. For example, in our study, the group of women with immigrant background was very heterogeneous, so

we cannot exclude that the increased antidepressant use is located on more specific country of birth categories. There is an ongoing debate whether these crude categorisations are feasible, and extra caution should be taken when investigating emerging intersectional categories rather than established ones.[66]

## Conclusion

It is important to recognise intersectional perspectives and interacting axes of oppression to tailor better public health interventions, as well as acknowledging the experiences of oppressed women to reach reproductive and social justice.[29 67] Our intersectional MAIHDA methodology operationalises this idea by providing information on the discriminatory accuracy of the contexts that define the intersectional strata. It highlights the need to consider disadvantages consisting of several interlocking structural dimensions such as income/class, age and immigration to better understand how HC might predispose certain women, mainly teenagers and low-income women with immigrant background, for depression. These vulnerabilities are based in inequalities that are not static, but structurally created and therefore possible to redeem.

**Author affiliations**
[1]Unit for Social Epidemiology, Department of Clinical Sciences, Lund University, Lund, Sweden
[2]Center for Multilevel Modelling, School of Education, University of Bristol, Bristol, UK
[3]Department of Gender Studies, Faculty of Social Sciences, Lund University, Lund, Sweden
[4]Center for Primary Health Care Research, Region Skåne, Region Skane Health Care, Malmö, Sweden

**Acknowledgements** A previous version of this study was presented as a poster at the Gynecological Endocrinology, the 19th World Congress in December 2020. We thank all colleagues at the Unit for Social epidemiology, Lund university, for valuable discussions.

**Contributors** SZ: conceptualisation, design, analysis, interpretation of data, writing original draft, final approvement of version to be published. KK and DM: interpretation of data, revising draft critically for intellectual content, final approvement of version to be published. RP-V: design, analysis, interpretation of data, revising draft critically for intellectual content, final approvement of version to be published. GL: analysis, interpretation of data, revising draft critically for intellectual content, final approvement of version to be published. JM: conceptualisation, design, analysis, interpretation of data, revising draft critically for intellectual content, final approvement of version to be published.

**Funding** This work was supported by The Swedish Research Council (Vetenskapsrådet) grant number: (2017-01321) https://www.swecris.se/betasearch/details/project/201701321VR.

**Competing interests** None declared.

**Patient consent for publication** Not required.

**Ethics approval** The database was approved by the Regional Ethical Review Board in Lund, Sweden, the Data Safety Board at Statistics Sweden and the National Board of Health and Welfare (Dnr: 2014/ 856, 2015/341).

**Provenance and peer review** Not commissioned; externally peer reviewed.

**Data availability statement** Public access to the data is restricted by the Swedish Authorities (Public Access to Information and Secrecy Act; http://www.government.se/information-material/2009/09/public-access-to-information-and-secrecy-act/) but data can be made available for researchers after a special review that includes approval of the research project by both an Ethics Committee and the authorities'

data safety committees. The National Board of Health and Welfare is a government agency under the Ministry of Health and Social Affairs. It is not their policy to provide individual level data to researchers abroad. Instead, they normally advise researchers in other countries to cooperate with Swedish colleagues, to whom they can provide data according to standard legal provisions and procedures. Requests for access to the data can be made to the National Board of Health and Welfare and Statistics Sweden (http://www.socialstyrelsen.se/statistics; https://www.scb.se/en/services/guidance-for-researchers-and-universities/).

**ORCID iDs**
Sofia Zettermark http://orcid.org/0000-0003-3181-8609
Juan Merlo http://orcid.org/0000-0001-8379-9708

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
