## [Reviewer comments · BMJ Open]

ARTICLE DETAILS

TITLE (PROVISIONAL)	Population Heterogeneity in Associations Between Hormonal Contraception and Antidepressant Use in Sweden: A Prospective Cohort Study Applying Intersectional Multilevel Analysis of Individual Heterogeneity and Discriminatory Accuracy (MAIHDA)
AUTHORS	Zettermark, Sofia; Khalaf, Kani; Perez-Vicente, Raquel; Leckie, George; Mulinari, Diana; Merlo, Juan

VERSION 1 – REVIEW

REVIEWER	Schaffir , Jonathan The Ohio State University
REVIEW RETURNED	31-Mar-2021

GENERAL COMMENTS	The authors analyze a national database of health records using a unique sociological statistics model to demonstrate susceptibilities to mood effects of prescribing hormonal contraception (HC). Their argument in brief is that sociological concerns such as age and immigrant status may contribute to the increased use of antidepressant medication that are seen in some users of HC. They do point out the many limitations of using such a database in their research, with antidepressants serving as a proxy for mental health issues and not a diagnosis, and the inability to demonstrate cause and effect. Even so, the conclusions are somewhat overstated with statements like "...consider disadvantages...to better understand how HC might predispose certain women...for depression." (p. 16), since they emphasize that HC may lead to depression. There are simply too many factors unaccounted for (indication for HC use, type of HC, dose, duration, compliance with prescription) to draw meaningful conclusions. Also, papers such as this lose sight of the fact that the main purpose for use of HC is the prevention of pregnancy, and unintended pregnancy itself may have more deleterious effects on mood and mental state than those discussed here. That said, I have the following questions: Why was the database divided primarily into two cohorts with and without a previous history of "mental health issues"? How is this term defined? Were there specific ICD-10 codes used? Did this include only mood disorders, or any psychiatric diagnosis? Which HC are included in the ATC codes described? Were IUDs included, which would not be expected to have systemic effects? The authors state that most prescriptions of HC are by midwives. Was there any attempt to determine the proportion prescribed by physicians? Would these be more likely to be prescribed for medical indications (abnormal bleeding patterns, dysmenorrhea, endometriosis, PMDD), which could also contribute to mental health? Including this point in the discussion would also be helpful in describing another limitation of the study.
--

REVIEWER	Roberts, Timothy University of Missouri, Pediatrics
REVIEW RETURNED	06-Apr-2021

GENERAL COMMENTS	Peer Review: Hormonal Contraception and Anti-depressant Use in Sweden: An Intersectional Multilevel Analysis of Individual Heterogeneity and Discriminatory Accuracy (MAIHDA) (bmjopen-2021-049553) Dr. Zettermark, Thank you for the opportunity to review this interesting study. As I understand it, this study is a secondary data analysis examining the influence of age, socioeconomic position, immigration status, and history of previous mental illness on the association between starting hormonal contraception and being prescribed an antidepressant over the subsequent year among women in Sweden. The large size of the database and completeness of the data available are strengths of this study. However, I have some concerns about how you and your team conceptualized the main exposure (use of hormonal contraction) and outcome (prescription of a SSRI) and included these items in the statistical analysis. I think addressing these concerns could improve your paper. Literature Review: This study examines impact of several biopsychosocial factors on the relationship between hormonal contraception and mood disorders. You correctly identify the contradictory evidence from randomized controlled trials finding both negative and positive influences of hormonal contraception on mood. You also quote two large population-based studies done in Nordic countries as evidence of a link between hormonal contraception and depression. However, I feel you need to be clearer that your study only found a link between hormonal contraception and depression among adolescents and not adults, unless I misunderstand your previous work: 7. Zettermark S, Perez Vicente R, Merlo J. Hormonal contraception increases the risk of psychotropic drug use in adolescent girls but not in adults: A pharmacoepidemiological study on 800 000 Swedish women. PLoS One 2018;13(3):e0194773 doi:10.1371/journal.pone.0194773. Also, you do not cite several more population-based studies that provide additional evidence, including a more recent study that found that the relationship between contraceptive use and SSRI use was confounded by health care utilization: Ditch S, Roberts TA, Hansen S. The influence of health care utilization on the association between hormonal contraception initiation and subsequent depression diagnosis and antidepressant use. Contraception. 2020 Apr 1;101(4):237-43. Please consider expanding your literature review and determining if any of these additional studies influence your study design or discussion. Risk Factor Exposure: As you and your team note, estrogen and progesterone are known to effect mood and modulate brain activity with subsequent mood
--

alterations in some women. The evidence for this effect is variable. Most researchers have examined the effect of hormonal contraceptives initiation as the change in hormone levels are responsible for the changes in mood. Also, this effect should build up over the first few weeks to months on the method and wear off over the first few weeks to months after stopping.

Your manuscript defined exposure to hormonal contraception in the following manner:

Users of HC were defined as any women who, according to the SPDR, filled a prescription of HC (Anatomical Therapeutic Chemical (ATC) classification system codes G02B, G03AAC) between 1 January 2010 and 31 December 2014, while non-users did not have a prescription filled during the same period.

You correctly excluded women with a history of pregnancy in the year before and after starting contraception as the hormonal shifts associated with pregnancy and the postpartum period are known risk factors for depression. However, I am concerned that you and your group did not exclude women who were on contraception prior to the study period. If hormonal changes associated with using hormonal contraception are linked to mood disorders, then a woman who is starting contraception for the first time is probably at higher risk than a woman who has been on the same contraceptive method for the last three years. A woman on a consistent hormonal method should have a relatively stable exposure to hormones. In a study examining contraceptive use among Swedish women from 2005-2010, found that 60.9% of women with a first prescription for contraception during the study period had been on contraception during the previous 18-months. [Josefsson A, Wiréhn A-B, Lindberg M, et al. Continuation rates of oral hormonal contraceptives in a cohort of first-time users: a population-based registry study, Sweden 2005–2010. *BMJ Open* 2013;3:e003401. doi:10.1136/bmjopen-2013-003401] The current study appears to examine a very similar population and may have a similar number percentage of women who are starting hormonal contraception for the first time and those who are continuing a prior prescription for contraception. I recommend excluding women who were on hormonal contraception at any time during the 1 to 1 1/2 year prior to their baseline date, the like the way your team excluded pregnant women or at least present the data for patients who are starting contraception for the first time separately from those who are continuing a prior prescription.

After identifying women who obtained a prescription for hormonal contraception, your team followed those women forward in time for one year to examine the risk of obtaining a prescription for a SSRI associated with being prescribed hormonal contraception. As you note in your discussion, your group cannot measure the actual use of the dispensed medications. This is an important factor when assessing the relationship between the exposure (hormonal contraception) and the outcome SSRI use. With the current definition of contraception use in this paper, a young woman who obtains a prescription for oral contraceptives and never takes them and a woman who has a subdermal implant in for a year are assumed to have the same risk of depression due to hormonal contraceptive use. In the study of contraceptive use among Swedish women I discussed above, 89.3% of the women were still on some form of hormonal contraception after 6-months and 78% were still on the same method (patient obtained a refill of the same method). [Josefsson A, Wiréhn A-B, Lindberg M, et al. Continuation rates of oral hormonal contraceptives in a cohort of first-time users: a population-based registry study, Sweden 2005–

2010. BMJ Open 2013;3:e003401. doi:10.1136/bmjopen-2013-003401] In a study conducted in the United States, the one-year continuation rates for oral contraceptives was 55%. [Peipert JF, Zhao Q, Allsworth JE, et al. Continuation and Satisfaction of Reversible Contraception, *Obstetrics & Gynecology* 2011;117:1105-1113 doi: 10.1097/AOG.0b013e31821188ad]. This suggests that a substantial number of the women in your study may discontinue use of hormonal contraception after starting and will have a declining exposure to the hormones, which is the major risk factor for this study. I suggest obtaining an estimate of continuation rates among patients starting contraception by examining how many obtained a refill of their method. Then either restrict your analysis to individuals who obtained a refill of medications or conduct your analysis with the whole sample and then with patients who had some evidence of taking the medications (i.e. obtained a refill).

A good example of this type of analysis is the study by Skovlund et al. that you quote in your paper. In this study the authors examined the risk associated with SSRI use among all women on hormonal contraception and the risk of depression diagnoses and first time SSRI use among women who started hormonal contraception for the first time during the study period. They found that the risk of depression and SSRI use associated with starting contraception increased during the first 6 months on a method and then decreased back down until the woman had been on a method for 12 months. Doing a similar sub-analysis on your data will make it easier to compare your data to previous studies.

Outcomes:

Your measurement of the exposure and the outcome in your study are dependent on patients accessing care. Women who come in to see a provider in the medical system and women who do not access medical care are different in multiple ways. Several time in your manuscript you quote articles describing barriers to care for different populations of women illustrating this point. In your study you are comparing a population of women who access care to obtain hormonal contraception to a population of women who are enrolled in the medical system but did not come in for contraception. Some of the women who did not come in for contraception may be just as likely or more likely to be seek care for medical issues as women who are seen for contraception. However, many of the women who are enrolled may never present for care for mood concerns or anything else. This difference in propensity to seek medical care for physical concerns such as contraception may also extend to the propensity of to seek medical care for emotional issues as well. As currently constructed, your study may be examining the propensity to seek care for mental health concerns among women who seek out medical care for contraception versus the propensity to seek care for mental health concerns among women who do not seek medical care for anything rather than an effect of hormonal contraception. A previous study of a United States population by Ditch et al. examined the risk of SSRI use and depression diagnoses among women starting hormonal contraception for the first time. [Ditch S, Roberts TA, Hansen S. The influence of health care utilization on the association between hormonal contraception initiation and subsequent depression diagnosis and antidepressant use. *Contraception*. 2020 Apr 1;101(4):237-43.] When comparing the group of women who started contraception to the population of

	women who were enrolled in the same healthcare system but did not start contraception, they found the same association of contraception with mood issues as had been found in prior studies. However, when the control group was restricted to women who had accessed the medical system at the same time, but had not started contraception, these associations disappeared. I recommend identifying a method to match your enrolled group and control group on frequency of accessing healthcare. This will allow you to better understand if the associations you are seeing between intersectional factors and the effect of hormonal contraception is due to an interaction between these factors and contraception use or an interaction between unmeasured factors and access to the healthcare system for reproductive and mental health concerns. I would be interested if this explains the differences you are seeing between the response of native women and immigrants in the response to hormonal contraception use.
--	--

REVIEWER	Lidegaard, Øjvind Rigshospitalet, University of Copenhagen, DK-2100 Copenhagen, Denmark, Gynecological Clinic 4232, DK-2100
REVIEW RETURNED	10-Apr-2021

GENERAL COMMENTS	This cross-sectional study examined around one million women 12-30 years old during a single year 2010 for the influence of several social indicators on the risk of using antidepressants when using hormonal contraception. The study demonstrated that young age, low income, and immigrants using HC had the largest relative risk of using antidepressants as compared to non-users of HC. Comments The strength of this study was the large number of women examined, providing a reasonable statistical power, the high number of social indicators examined, and the relatively advanced statistical analysis. There were, however, also important limitations. First the cross-sectional design did not permit to follow a cohort of women beginning using HC. The study chose not-current users of HC, in the paper called non-users as reference group. This reference group include all the former-users of HC who had stopped using HC due to mood-deterioration or regular depression development (more than three years previously). Many of these women will have a long-term risk of depression development – not due to their previous use of HC – but that former use was the first “test” of their sensitivity towards depression development. This point is illustrated in a large Danish study (ref. 6). COC implied with non-users a relative risk of antidepressant use of 1.2 (1.22-1.25)(Table 2) and with never-users as reference of 2.2 (2.18-2.31)(suppl. Table 4). The overall risk estimate was thus increased six-fold with this change in reference group. For rare diseases such as thrombosis risk, it doesn't matter which reference group you choose, but for frequent outcomes such as mood changes with HC use it is a crucial circumstance. The stratification of women who within the latest three years had mental disease, partly control for this bias, but mental disease more than three years ago could still indicate a susceptibility for later mental disease.
---

	It also makes a huge difference in risk estimates whether you follow women from they begin using HC or you assess the risk in women several years after having started on HC. In the latter situation a large proportion of those experiencing depressive symptoms have ceased by using HC and has left a “healthy still user cohort”, tolerating HC well. By making both fallacies in a crossectional study with a frequent outcome, the risk of depression could be severely underestimated. The risk of depression in the numerator is severely underestimated and the risk of depression in the denominator severely over-estimated. Thereby – even when all the other factors are taking into account – the risk estimates of antidepressant use will generally be underestimated. Unfortunately, it is not possible in a crossectional study to change these two important epidemiological circumstances, but it should be acknowledged in the discussion section. This important methodological circumstance does necessarily invalidate the multilevel analysis of individual heterogeneity and discriminatory accuracy analysis which is the main focus of this study. But a cohort design would have been much stronger because it could identify never-users of HC to be used as the reference group instead of non-users, and data for such a historical prospective analysis is available in the Swedish registers. The multilevel analysis of individual heterogeneity and discriminatory accuracy analysis is interesting and deserves publication, as such analyses are rarely conducted. The method description is not very easy to follow – even for a skilled epidemiologist – but seems sound, and I am not sure it could have been done simpler. The problem is that few clinicians are familiar with MAIHDA analyses. On page 14 bottom the authors state: “The fact that adult women native to Sweden were almost unaffected by HC use, could strengthen this suggestion. Without the intersectional strata this disparity would not have been so easily identified and visualized”. This statement could be biased by the fact that adult native Swedish women are likely to be long-term users of HC, a selected group of women tolerating HC well, because those sensitive for HC had left this cohort years earlier. This potential but likely bias should be acknowledged. In conclusion the classical epidemiological part of the study (the crossectional design) limits the validity of the results of the advanced MAIHDA approach.
--	---

VERSION 1 – AUTHOR RESPONSE

Reviewer: 1

Dr. Jonathan Schaffir , The Ohio State University

Comments to the Author:

The authors analyze a national database of health records using a unique sociological statistics model to demonstrate susceptibilities to mood effects of prescribing hormonal contraception (HC). Their argument in brief is that sociological concerns such as age and immigrant status may contribute to the increased use of antidepressant medication that are seen in some users of HC. They do point

out the many limitations of using such a database in their research, with antidepressants serving as a proxy for mental health issues and not a diagnosis, and the inability to demonstrate cause and effect.,

Dr Schaffir, thank you for your succinct review with many insights that will help improve our study. Below is our answers to your questions and concerns, point by point.

1. Even so, the conclusions are somewhat overstated with statements like "...consider disadvantages...to better understand how HC might predispose certain women...for depression." (p. 16), since they emphasize that HC may lead to depression.

The reviewer is correct in that causal effects cannot be concluded from this study since it is observational in nature. Epidemiological studies like ours can however contribute to a better understanding of patterns of drug utilization and the diseases/symptoms they are used for. Exploring heterogeneity in known associations through a MAIHDA approach could contribute to both guidance in further clinical studies and theoretical/methodological development.

We thank the reviewer for pointing out that certain phrasing may previously implied causal links rather than emphasizing population heterogeneity, and have remedied that in our revised manuscript.

2. There are simply too many factors unaccounted for (indication for HC use, type of HC, dose, duration, compliance with prescription) to draw meaningful conclusions.

Unmeasured confounding and bias is a challenge in all observational studies. We thank the reviewer for his suggestions and agree that many factors are unaccounted for in our register-based study, but we argue that the results are still of interest. The existence of a potential causal association between HC use and impaired mental health has been suggested in several previous studies, see for example:

- Skovlund CW, Mørch LS, Kessing LV, et al. Association of Hormonal Contraception With Depression. *JAMA Psychiat* 2016;73(11):1154-62
- Zettermark S, Perez Vicente R, Merlo J. Hormonal contraception increases the risk of psychotropic drug use in adolescent girls but not in adults: A pharmacoepidemiological study on 800 000 Swedish women. *PLoS One* 2018;13(3):e0194773
- Zethraeus N, Dreber A, Ranehill E, et al. A first-choice combined oral contraceptive influences general well-being in healthy women: a double-blind, randomized, placebo-controlled trial. *Fertil Steril* 2017;107(5):1238-45.
- Lundin C, Malmborg A, Slezak J, et al. Sexual function and combined oral contraceptives - a randomised, placebo-controlled trial. *Endocr Connect* 2018;7(11):1208-1216
- de Wit AE, Booij SH, Giltay EJ, et al. Association of Use of Oral Contraceptives With Depressive Symptoms Among Adolescents and Young Women. *JAMA Psychiat* 2020;77(1):52-59

Against this background, our approach did not pretend to further prove causal effects. Our main research question was to provide an extended epidemiological description of the association between HC and antidepressant use. We do so by using a prospective cohort study design as the women are followed for antidepressant use once their exposure to HC has been identified. The MAIHDA allows us to compare vis-a-vis strata of women with similar characteristics that only differ in their use of HC. In this way we provided an improved mapping of the use of antidepressants in relation to HC in young women.

We thereby aimed to explore the heterogeneity that exists in the already observed associations between HC and depressive symptoms, whereby confounding and bias must be addressed and discussed but does not necessarily invalidate the conclusions. Nevertheless, the

extended stratification included in the MAIHDA approach adjusts by design at least for the variables defining the intersectional strata.

Bearing that in mind, we understand that possible confounding and bias need to be further addressed in our study and we did a separate analysis investigating the length of use, where new users were defined as not having any HC prescription during the last 3 years. New users made up 26.2% of the total number of HC users (11.1% of the total cohort). We ran a logistic regression with anti-depressants as the outcome to control for differences between the groups (new users, continuous users and non-users) with non-users as the reference group, where the results were very similar between the HC user-groups: OR 1.45 (95% CI 1.41-1.50) for continuous users and OR 1.52 (1.46-1.60) for new users. This led us to conclude that there is no major influence of user-time on our results. A short discussion about it has been added in the discussion part of the revised manuscript.

We would also like to point out that our data are not based on issued prescriptions, but on dispensed prescriptions at pharmacies, meaning we know the woman received this particular prescription, making intake of the drug more likely, although still not possible to determine without doubt.

3. Also, papers such as this lose sight of the fact that the main purpose for use of HC is the prevention of pregnancy, and unintended pregnancy itself may have more deleterious effects on mood and mental state than those discussed here.

We thank the reviewer for his insightful observation and have added a sentence pointing this out to further nuance the discussion. We do not argue that the side effects of HC some women experience merits a general reduction in utilization, since the benefits of safe pregnancy protection is crucial to many aspects of reproductive health and bodily autonomy. However, compliance is known to be low and discontinuation due to mental health side effects high, rendering an investigation of heterogeneity in response important to later on be able to address these factors and increase compliance and satisfaction with HC.

4. Why was the database divided primarily into two cohorts with and without a previous history of "mental health issues"? How is this term defined? Were there specific ICD-10 codes used? Did this include only mood disorders, or any psychiatric diagnosis?

The specific ICD-10 codes used (F00-F99) were indicated in the originally submitted manuscript (page 6, line 163-164). In the revised manuscript the definition of mental health issues reads: "Previous mental health issues were defined as having any psychiatric disorder diagnosed at a hospital (ICD: F00-F99) or a dispensed prescription of a psychotropic drug (ATC: N05A, N05B, N06A) in the past three years."

The stratification on previous mental health issues was made since women with a history of mental health disorders have an appreciably higher risk of developing these again. Previous studies have pointed out that these women may be more susceptible to adverse effects of HC. See for example

- Bengtsdotter H, Lundin C, Gemzell Danielsson K, Bixo M, Baumgart J, Marions L, et al. Ongoing or previous mental disorders predispose to adverse mood reporting during combined oral contraceptive use. *Eur J Contracept Repr* 2018;23(1):45-51
- Worly BL, Gur TL, Schaffir J. The relationship between progestin hormonal contraception and depression: a systematic review. *Contraception* 2018;97(6):478-89

The decision to stratify and investigate rather than exclude these women was therefore motivated. Another possible approach would have been to include “mental health” as a dimension in the intersectional strata, but from the perspective of intersectional theory, stratification is more coherent as mental health is not properly a power dimension as such.

5. Which HC are included in the ATC codes described? Were IUDs included, which would not be expected to have systemic effects?

As described on page 6 line 154-156 in the original manuscript the ATC codes were G03A (hormonal contraceptives for systemic use) excluding G03AD (emergency contraceptives) and G02B (hormonal contraceptives for external use), including all the subclassification for those codes according to the ATC-tree: G03AA (progesterone and estrogen), G03AB (progesterone and estrogen, sequence formulas), G03AC (progesterone), G02BA (intrauterine) and G02BB (intravaginal). Frequency tables has been added and can be viewed in Supplementary material 6.

IUDs has been included (ATC G02BA), and make up 3.25% of the total number of HC prescriptions. Although systemic effects are regarded as less likely, systemic uptake and side effects such as headache, acne and mood effects are known to occur in IUD users as well, motivating their inclusion. See for example:

- Kailasam C, Cahill D. Review of the safety, efficacy and patient acceptability of the levonorgestrel-releasing intrauterine system. *Patient Prefer Adherence*. 2008;2:293–302 PubMed .
6. The authors state that most prescriptions of HC are by midwives. Was there any attempt to determine the proportion prescribed by physicians? Would these be more likely to be prescribed for medical indications (abnormal bleeding patterns, dysmenorrhea, endometriosis, PMDD), which could also contribute to mental health? Including this point in the discussion would also be helpful in describing another limitation of the study.

As the reviewer highlights it is possible that prescriptions issued by physicians are more likely to be prescribed for medical rather than contraceptive purposes, which might lead to confounding by indication (e.g., PMDD causes impaired psychological health). This situation does not exclude an adverse effect on mental health and, as, explained above, our study is explorative. Nevertheless, the comment of the referee points to differences within the population that we have not unaccounted for. We therefore re-did the analysis excluding 62 807 women with prescriptions issued by physicians (13.4%) or by an unknown issuer (0.57%). The remaining 86.0% of the total prescriptions were issued by midwives.. The results from the new analyses were very similar and they do not affect our conclusions in an appreciable way. However, the revised manuscript is based on this new population.

Reviewer: 2

Dr. Timothy Roberts, University of Missouri

Comments to the Author:

Peer Review: Hormonal Contraception and Anti-depressant Use in Sweden: An Intersectional Multilevel Analysis of Individual Heterogeneity and Discriminatory Accuracy (MAIHDA) (bmjopen-2021-049553)

Dr. Zettermark,

Thank you for the opportunity to review this interesting study. As I understand it, this study is a secondary data analysis examining the influence of age, socioeconomic position, immigration status,

and history of previous mental illness on the association between starting hormonal contraception and being prescribed an antidepressant over the subsequent year among women in Sweden. The large size of the database and completeness of the data available are strengths of this study. However, I have some concerns about how you and your team conceptualized the main exposure (use of hormonal contraception) and outcome (prescription of a SSRI) and included these items in the statistical analysis. I think addressing these concerns could improve your paper.

Dr Roberts, thank you for your substantial review with many insights that will help improve our study. Below is our answers to your questions and concerns, point by point.

1. Literature Review:

This study examines impact of several biopsychosocial factors on the relationship between hormonal contraception and mood disorders. You correctly identify the contradictory evidence from randomized controlled trials finding both negative and positive influences of hormonal contraception on mood. You also quote two large population-based studies done in Nordic countries as evidence of a link between hormonal contraception and depression. However, I feel you need to be clearer that your study only found a link between hormonal contraception and depression among adolescents and not adults, unless I misunderstand your previous work: 7. Zettermark S, Perez Vicente R, Merlo J. Hormonal contraception increases the risk of psychotropic drug use in adolescent girls but not in adults: A pharmacoepidemiological study on 800 000 Swedish women. *PLoS One* 2018;13(3):e0194773 doi:10.1371/journal.pone.0194773. Also, you do not cite several more population-based studies that provide additional evidence, including a more recent study that found that the relationship between contraceptive use and SSRI use was confounded by health care utilization: Ditch S, Roberts TA, Hansen S. The influence of health care utilization on the association between hormonal contraception initiation and subsequent depression diagnosis and antidepressant use. *Contraception*. 2020 Apr 1;101(4):237-43. Please consider expanding your literature review and determining if any of these additional studies influence your study design or discussion.

Thank you for discussing the literature review. The reviewer is correct in that the association between HC and psychotropic drugs was only present in adolescents in our previous study. This has been expressed more clearly in the revised manuscript. It is always a balancing act to include enough previous studies in the literature review to provide a sound background for one's hypothesis while keeping the discussion succinct and relevant. A few more recent studies as well as the references provided by the referee have now been included in the discussion part of the revised manuscript. The issue of confounding by health care utilization is discussed further down.

2. Risk Factor Exposure:

As you and your team note, estrogen and progesterone are known to effect mood and modulate brain activity with subsequent mood alterations in some women. The evidence for this effect is variable. Most researchers have examined the effect of hormonal contraceptives initiation as the change in hormone levels are responsible for the changes in mood. Also, this effect should build up over the first few weeks to months on the method and wear off over the first few weeks to months after stopping.

Your manuscript defined exposure to hormonal contraception in the following manner:

Users of HC were defined as any women who, according the SPDR, filled a prescription of HC (Anatomical Therapeutic Chemical (ATC) classification system codes G02B, G03AA-C) between 1 January 2010 and 31 December 2014, while non-users did not have a prescription filled during the same period. You correctly excluded women with a history of pregnancy in

the year before and after starting contraception as the hormonal shifts associated with pregnancy and the postpartum period are known risk factors for depression. However, I am concerned that you and your group did not exclude women who were on contraception prior to the study period. If hormonal changes associated with using hormonal contraception are linked to mood disorders, then a woman who is starting contraception for the first time is probably at higher risk than a woman who has been on the same contraceptive method for the last three years. A woman on a consistent hormonal method should have a relatively stable exposure to hormones. In a study examining contraceptive use among Swedish women from 2005-2010, found that 60.9% of women with a first prescription for contraception during the study period had been on contraception during the previous 18-months. [Josefsson A, Wiréhn A-B, Lindberg M, et al. Continuation rates of oral hormonal contraceptives in a cohort of first-time users: a population-based registry study, Sweden 2005–2010. *BMJ Open* 2013;3:e003401. doi:10.1136/bmjopen-2013-003401] The current study appears to examine a very similar population and may have a similar number percentage of women who are starting hormonal contraception for the first time and those who are continuing a prior prescription for contraception. I recommend excluding women who were on hormonal contraception at any time during the 1 to 1 1/2 year prior to their baseline date, the like the way your team excluded pregnant women or at least present the data for patients who are starting contraception for the first time separately from those who are continuing a prior prescription.

Following the reviewer's suggestion we did a separate analysis investigating the length of use, where new users were defined as not having any HC prescription during the last 3 years. New users made up 26.2% of the total number of HC users (11.1% of the total cohort). We ran a logistic regression with anti-depressants as the outcome to control for differences between the groups (new users, continuous users and non-users) with non-users as the reference group, where the results were very similar between the HC user-groups: OR 1.45 (95% CI 1.41-1.50) for continuous users and OR 1.52 (1.46-1.60) for new users. This led us to conclude that there is no a major influence of user-time on our study. A short comment on this issue has been added to the discussion part of the revised manuscript.

3. After identifying women who obtained a prescription for hormonal contraception, your team followed those women forward in time for one year to examine the risk of obtaining a prescription for a SSRI associated with being prescribed hormonal contraception. As you note in your discussion, your group cannot measure the actual use of the dispensed medications. This is an important factor when assessing the relationship between the exposure (hormonal contraception) and the outcome SSRI use. With the current definition of contraception use in this paper, a young woman who obtains a prescription for oral contraceptives and never takes them and a woman who has a subdermal implant in for a year are assumed to have the same risk of depression due to hormonal contraceptive use.

This is indeed a limitation in our register-based study, and ideally a measurement of actual drug intake would have been available. That would however require a completely different design with for example interviewers, making an investigation of hundreds of thousands of women very difficult. We will have to assume that most women who were dispensed a prescription at a pharmacy also take the medication. However, in presence of misclassification of the exposure (exposed women according to the register are actually not exposed because they do not take the dispensed HC) we would find a bias towards the null that underestimate the true association between HC and antidepressant. Thus, the associations we present might be viewed as conservative. In any case, the existence of a potential causal association between HC use and impaired psychological health has been suggested in several previous studies, see for example:

- Skovlund CW, Morch LS, Kessing LV, et al. Association of Hormonal Contraception With Depression. *JAMA Psychiat* 2016;73(11):1154-62
- Zettermark S, Perez Vicente R, Merlo J. Hormonal contraception increases the risk of psychotropic drug use in adolescent girls but not in adults: A pharmacoepidemiological study on 800 000 Swedish women. *PLoS One* 2018;13(3):e0194773
- Zethraeus N, Dreber A, Ranehill E, et al. A first-choice combined oral contraceptive influences general well-being in healthy women: a double-blind, randomized, placebo-controlled trial. *Fertil Steril* 2017;107(5):1238-45.
- Lundin C, Malmborg A, Slezak J, et al. Sexual function and combined oral contraceptives - a randomised, placebo-controlled trial. *Endocr Connect* 2018;7(11):1208-1216
- de Wit AE, Booij SH, Giltay EJ, et al. Association of Use of Oral Contraceptives With Depressive Symptoms Among Adolescents and Young Women. *JAMA Psychiat* 2020;77(1):52-59).

Against this background, our approach did not pretend to further prove causal effects but rather points out that heterogeneity exists in the already observed associations between HC and depressive symptoms, whereby confounding and bias must be addressed and discussed but where these issues do not necessarily invalidate the conclusions.

4. In the study of contraceptive use among Swedish women I discussed above, 89.3% of the women were still on some form of hormonal contraception after 6-months and 78% were still on the same method (patient obtained a refill of the same method). [Josefsson A, Wiréhn A-B, Lindberg M, et al. Continuation rates of oral hormonal contraceptives in a cohort of first-time users: a population-based registry study, Sweden 2005–2010. *BMJ Open* 2013;3:e003401. doi:10.1136/bmjopen-2013-003401] In a study conducted in the United States, the one-year continuation rates for oral contraceptives was 55%. [Peipert JF, Zhao Q, Allsworth JE, et al. Continuation and Satisfaction of Reversible Contraception, *Obstetrics & Gynecology* 2011;117:1105-1113 PubMed doi: 10.1097/AOG.0b013e31821188ad]. This suggests that a substantial number of the women in your study may discontinue use of hormonal contraception after starting and will have a declining exposure to the hormones, which is the major risk factor for this study. I suggest obtaining an estimate of continuation rates among patients starting contraception by examining how many obtained a refill of their method. Then either restrict your analysis to individuals who obtained a refill of medications or conduct your analysis with the whole sample and then with patients who had some evidence of taking the medications (i.e. obtained a refill). A good example of this type of analysis is the study by Skovlund et al. that you quote in your paper. In this study the authors examined the risk associated with SSRI use among all women on hormonal contraception and the risk of depression diagnoses and first time SSRI use among women who started hormonal contraception for the first time during the study period. They found that the risk of depression and SSRI use associated with starting contraception increased during the first 6 months on a method and then decreased back down until the woman had been on a method for 12 months. Doing a similar sub-analysis on your data will make it easier to compare your data to previous studies.

Thank you for this reflection. In the revised manuscript we have included the references provided by the referee and also clarified that the existence of a potential causal association between HC use and impaired psychological health has been suggested in several previous studies. We have also added a section in the limitations section where we address this limitation.

However, we also stress that our approach did not pretend to further prove causal effects but rather pointing out that heterogeneity exists in the already observed associations between HC and

depressive symptoms, whereby confounding and bias must be addressed and discussed but does not necessarily invalidate the conclusions. See also our answers above.

5. Outcomes:

Your measurement of the exposure and the outcome in your study are dependent on patients accessing care. Women who come in to see a provider in the medical system and women who do not access medical care are different in multiple ways. Several times in your manuscript you quote articles describing barriers to care for different populations of women illustrating this point. In your study you are comparing a population of women who access care to obtain hormonal contraception to a population of women who are enrolled in the medical system but did not come in for contraception. Some of the women who did not come in for contraception may be just as likely or more likely to seek care for medical issues as women who are seen for contraception. However, many of the women who are enrolled may never present for care for mood concerns or anything else. This difference in propensity to seek medical care for physical concerns such as contraception may also extend to the propensity of to seek medical care for emotional issues as well. As currently constructed, your study may be examining the propensity to seek care for mental health concerns among women who seek out medical care for contraception versus the propensity to seek care for mental health concerns among women who do not seek medical care for anything rather than an effect of hormonal contraception. A previous study of a United States population by Ditch et al. examined the risk of SSRI use and depression diagnoses among women starting hormonal contraception for the first time. [Ditch S, Roberts TA, Hansen S. The influence of health care utilization on the association between hormonal contraception initiation and subsequent depression diagnosis and antidepressant use. *Contraception*. 2020 Apr 1;101(4):237-43.] When comparing the group of women who started contraception to the population of women who were enrolled in the same healthcare system but did not start contraception, they found the same association of contraception with mood issues as had been found in prior studies. However, when the control group was restricted to women who had accessed the medical system at the same time, but had not started contraception, these associations disappeared. I recommend identifying a method to match your enrolled group and control group on frequency of accessing healthcare. This will allow you to better understand if the associations you are seeing between intersectional factors and the effect of hormonal contraception is due to an interaction between these factors and contraception use or an interaction between unmeasured factors and access to the healthcare system for reproductive and mental health concerns. I would be interested if this explains the differences you are seeing between the response of native women and immigrants in the response to hormonal contraception use.

In Sweden, health care is universal for all citizens (i.e., no enrollment or insurance needed) and visits to midwives for contraceptive counselling is free of charge for the patient (a visit to a doctor costs approximately 15 US dollars, and is free for anyone under 18), so access for our population is good. However, utilization can still differ among population groups as the reviewer points out. To investigate the hypothesis of access to care we ran a sensitivity analysis excluding women who had not been in contact with the health care system for the last 3 years. We operationalized health care utilization as having received any diagnosis at a hospital (in- and outpatient) or any prescription at the primary health care in the last 3 years.

Unfortunately we do not have access to data on diagnosis within the primary care, only prescriptions. Even if many primary care contacts lead to prescriptions, the true number of health care contacts will therefore be higher than our operationalization shows (a visit to a primary care center without receiving a prescription will not be registered as a contact in our definition). In addition, following a previous reviewer's suggestion we also excluded women

who received a HC prescription from a physician rather than from a midwife since HC prescriptions from physicians may be in response to other medical conditions rather than for contraception purposes. In contrast, midwives are only allowed to prescribe HC for contraceptive purposes. This reduced our cohort to 915 954 women and the care-accessors with this definition was 60,5%. The sensitivity analysis including only these women (i.e., care-accessors with a midwife prescription of HC) is attached as Supplementary material 7 and shows no major differences in the results. Adolescent girls and low-income immigrant women on HC still have the highest differences in anti-depressant use compared to their non-HC using counterparts. This is discussed in the revised manuscript.

Reviewer: 3

Prof. Øjvind Lidegaard, Rigshospitalet, University of Copenhagen, DK-2100 Copenhagen, Denmark

Prof. Lidegaard, thank you for your observant review with many insights that will help improve our study. Below are our answers to your questions and concerns, point by point.

Comments to the Author:

1. This cross-sectional study examined around one million women 12-30 years old during a single year 2010 for the influence of several social indicators on the risk of using antidepressants when using hormonal contraception. The study demonstrated that young age, low income, and immigrants using HC had the largest relative risk of using antidepressants as compared to non-users of HC.

Two parts of your summary "This cross-sectional study" and "during a single year 2010" have led us to clarify our study design both here and in the paper. Our study has a short but complete follow-up period of one year for each individual woman. The baseline corresponds to date for the exposure (HC) and thereafter we follow the women for the outcome (anti-depressants) within one year from the baseline date. We would therefore not consider our study as a cross-sectional design where the temporality of exposure and outcome is unknown, but a longitudinal study with a short (and complete) follow-up. The original cohort is based on population data drawn for the year 2010, but the individual baseline date is assigned to each woman based on her first HC-prescription any time during 2010-2014. A woman obtaining her first prescription 1st of September 2013 is therefore followed to the 1st of September 2014. For non-users of HC, the baseline date could not be based on a HC-prescription and was therefore assigned, to 1st of July 2012. This means all non-users had been true non-users for at least 1.5 years before their follow-up started (1st January 2010 to 1st July 2012) but also continued to be non-users all the way to 31st dec 2014. We have clarified the study design in the methods section of the revised manuscript,

2. The strength of this study was the large number of women examined, providing a reasonable statistical power, the high number of social indicators examined, and the relatively advanced statistical analysis.

There were, however, also important limitations. First the cross-sectional design did not permit to follow a cohort of women beginning using HC. The study chose not-current users of HC, in the paper called non-users as reference group. This reference group include all the former-users of HC who had stopped using HC due to mood-deterioration or regular depression development (more than three years previously). Many of these women will have a long-term risk of depression development – not due to their previous use of HC – but that former use was the first "test" of their sensitivity towards depression development. This point is illustrated in a large Danish study (ref. 6). COC implied with non-users a relative risk of antidepressant use of 1.2 (1.22-1.25)(Table 2) and with never-users as reference of 2.2 (2.18-2.31)(suppl.

Table 4). The overall risk estimate was thus increased six-fold with this change in reference group. For rare diseases such as thrombosis risk, it doesn't matter which reference group you choose, but for frequent outcomes such as mood changes with HC use it is a crucial circumstance. The stratification of women who within the latest three years had mental disease, partly control for this bias, but mental disease more than three years ago could still indicate a susceptibility for later mental disease. It also makes a huge difference in risk estimates whether you follow women from they begin using HC or you assess the risk in women several years after having started on HC. In the latter situation a large proportion of those experiencing depressive symptoms have ceased by using HC and has left a "healthy still user cohort", tolerating HC well.

The reviewer points to important limitations, but as explained above our study design is not cross-sectional. We agree that there is a risk of measuring a "healthy still user cohort", which we did reason around in the introduction and discussion of the original manuscripts as a "healthy survivor" effect or "selective discontinuation bias", but have now expanded on the revised manuscript.

To further address the issue raised by the reviewer, as the HC users in our cohort could have obtained a prescription before baseline in 2010 as well, we did a separate analysis investigating the length of use. New users were defined as not having any HC prescription during the last 3 years (counted from their own baseline date). New users made up 26.2% of the total number of HC users (11.1% of the total cohort). We ran a logistic regression with anti-depressants as the outcome to control for differences between the groups (new users, continuous users and non-users) with non-users as the reference group, where the results were very similar between the HC user-groups: OR 1.45 (95% CI 1.41-1.50) for continuous users and OR 1.52 (1.46-1.60) for new users. This led us to conclude that there is no major influence of user-time on our results. A short discussion about it has been added in the discussion part of the revised manuscript.

3. By making both fallacies in a cross-sectional study with a frequent outcome, the risk of depression could be severely underestimated. The risk of depression in the numerator is severely underestimated and the risk of depression in the denominator severely over-estimated. Thereby – even when all the other factors are taken into account – the risk estimates of antidepressant use will generally be underestimated. Unfortunately, it is not possible in a cross-sectional study to change these two important epidemiological circumstances, but it should be acknowledged in the discussion section. This important methodological circumstance does necessarily invalidate the multilevel analysis of individual heterogeneity and discriminatory accuracy analysis which is the main focus of this study. But a cohort design would have been much stronger because it could identify never-users of HC to be used as the reference group instead of non-users, and data for such a historical prospective analysis is available in the Swedish registers.

Thank you for this interesting insight. We have indicated in the discussion part that our results may underestimate the association between HC and antidepressant use. While this is not ideal, the issue of underestimation does not challenge the soundness of the conclusions per se. We have also aimed to limit possible bias by several approaches as explained above. Unmeasured bias is a challenge in all observational studies. We thank the reviewer for his suggestions and agree that many factors are unaccounted for in our register based study, but this does not invalidate register based epidemiology. Our approach is not apt to prove causal effects but rather points out that heterogeneity exists in the already observed associations between HC and depressive symptoms.

Furthermore, causal effects cannot be concluded from this study despite the changes suggested, since it's still observational in nature. Epidemiological studies like ours can however contribute to a better understanding of population level patterns of drug utilization and the diseases/symptoms they

are used for treating, such as depression or anxiety. Exploring heterogeneity in known associations such as that between HC and adverse mental health outcomes where many women do not react negatively, but some do through a MAIHDA approach could contribute to both guidance in further clinical studies and theoretical/methodological development. However, we realize that some of our expressions in the original manuscript may suggest we aimed to detect a causal association and have remedied that in our revised manuscript.

4. The multilevel analysis of individual heterogeneity and discriminatory accuracy analysis is interesting and deserves publication, as such analyses are rarely conducted. The method description is not very easy to follow – even for a skilled epidemiologist – but seems sound, and I am not sure it could have been done simpler. The problem is that few clinicians are familiar with MAIHDA analyses.

Thank you very much for this opinion. We agree the methodology is new and we are making efforts to communicate it outside the sociological field. The present manuscript is relevant for helping to introduce MAIHDA within a more clinical (epidemiological) context. For further reading on the methodology we can recommend a few papers:

- Evans CR, Williams DR, Onnela JP, et al. A multilevel approach to modeling health inequalities at the intersection of multiple social identities. *Soc Sci Med* 2018;203:64-73 PubMed .
 - Bauer GR. Incorporating intersectionality theory into population health research methodology: Challenges and the potential to advance health equity. *Soc Sci Med* 2014;110:10–7 PubMed .
 - Wemrell M, Mulinari S, Merlo J. An intersectional approach to multilevel analysis of individual heterogeneity (MAIH) and discriminatory accuracy. *Soc Sci Med* 2017;178:217-9 PubMed .
 - Merlo J. Multilevel analysis of individual heterogeneity and discriminatory accuracy (MAIHDA) within an intersectional framework. *Soc Sci Med* 2018;203:74-80 PubMed .
5. On page 14 bottom the authors state: “The fact that adult women native to Sweden were almost unaffected by HC use, could strengthen this suggestion. Without the intersectional strata this disparity would not have been so easily identified and visualized”. This statement could be biased by the fact that adult native Swedish women are likely to be long-term users of HC, a selected group of women tolerating HC well, because those sensitive for HC had left this cohort years earlier. This potential but likely bias should be acknowledged.

Thank you for this observation which is in line with opinions of other referees (see above). As indicated above, we have also performed a sensitivity analysis investigating the length of HC use, and we conclude that there is not a major influence of user-time on our results. The results from the sensitivity analysis is attached and a short discussion about it has been added in the discussion part of the revised manuscript.

6. In conclusion the classical epidemiological part of the study (the cross-sectional design) limits the validity of the results of the advanced MAIHDA approach.

As explained above, our study is not cross-sectional but longitudinal with a short individual follow up of equal length for all the women. Furthermore we have done sensitivity analysis following the reviewer’s suggestion to explore the issue of length of HC use. As in any other observational study proving causal effects is difficult in MAIHDA analysis. However, the aim of our study was not to prove causal links but to display population heterogeneity in a parsimonious and theoretically sound way, which we believe our study contributes with.

VERSION 2 – REVIEW

REVIEWER	Roberts, Timothy University of Missouri, Pediatrics
REVIEW RETURNED	18-Jun-2021

GENERAL COMMENTS	Thank you for your thoughtful revision of your paper and response to our suggestions. You have addressed the majority of my concerns. I still have concerns about the primary outcome in your study. Several of the studies you quote suggest that there is an increase in the risk of depression with changes in hormones associated with starting hormonal contraception. For example, Skovlund et al. (2016) found that the risk of depression associated with starting OCP peaked at 6-months and then slowly came down to baseline with continued use. In the current study you combined current users and new starts together as contraception users and used women who were not on contraception during the study period as the control group. In your response to reviewer concerns, you compared the risk of depression between new users and current users and found no difference. This is contrary to the previous literature you quote in your introduction. This makes me concerned about your study outcome. In your prior study using a similar study population, Zettermark (2018), you addressed this issue by excluding patients who had used contraception during the prior 4 years and compared women who started hormonal contraception during the one year study period versus women who did not. I wonder why you did not use this definition in your current study. Prof. Øjvind Lidegaard had similar concerns.
--

REVIEWER	Lidegaard, Øjvind Rigshospitalet, University of Copenhagen, DK-2100 Copenhagen, Denmark, Gynecological Clinic 4232, DK-2100
REVIEW RETURNED	11-Jul-2021

GENERAL COMMENTS	I think the authors have addressed and clarified the concerns expressed in my primary review of the paper. The issue is important, and the study provides new insight into the complexity of factors influencing use of antidepressants with use of hormonal contraception. Appropriate sensitivity analyses were made to address the issue of length of use, and the design is now clearly explained as being a historical prospective cohort study. Can be published as it is.
---

VERSION 2 – AUTHOR RESPONSE

Reviewer: 2

Dr. Timothy Roberts, University of Missouri

Comments to the Author:

#1 Thank you for your thoughtful revision of your paper and response to our suggestions. You have addressed the majority of my concerns.

We are glad the majority of your concerns are addressed. Thank you for your help in improving our paper.

#2 I still have concerns about the primary outcome in your study. Several of the studies you quote suggest that there is an increase in the risk of depression with changes in hormones associated with starting hormonal contraception. For example, Skovlund et al. (2016) found that the risk of depression associated with starting OCP peaked at 6-months and then slowly came down to baseline with continued use. In the current study you combined current users and new starts together as contraception users and used women who were not on contraception during the study period as the control group. In your response to reviewer concerns, you compared the risk of depression between new users and current users and found no difference. This is contrary to the previous literature you quote in your introduction. This makes me concerned about your study outcome. In your prior study using a similar study population, Zettermark (2018), you addressed this issue by excluding patients who had used contraception during the prior 4 years and compared women who started hormonal contraception during the one year study period versus women who did not. I wonder why you did not use this definition in your current study.

We thank the reviewer for further expanding on the issue of new versus continuous HC users as it is important in understanding the possible mechanism for adverse mental health outcomes. However, our approach does not pretend to further prove causal effects but rather to explore heterogeneity. Our main research question was to provide an extended epidemiological description of the association between HC and antidepressant use. The MAIHDA allows us to compare vis-a-vis strata of women with similar characteristics that only differ in their use of HC. In this way we provided an improved mapping of the use of antidepressants in relation to HC in young women. We thereby aimed to explore the heterogeneity that exists in the already observed associations between HC and depressive symptoms.

We realize we made a mistake in our wording responding to previous reviewer concerns. When comparing the antidepressant use between new users and current users we did find a small difference in the point estimations, although the 95% confidence intervals overlapped. New users had a slightly higher risk than non-users which is in agreement with previous results. This has been corrected in the revised manuscript.

We have also performed another sensitivity analysis, excluding all users who had a prescription fill within the past five years from their baseline (past and continuous users) as suggested by the reviewer. We then reran the MAIHDA with only new users (no past HC use) and never-users as the reference group. Overall, the results indicate a slightly higher absolute antidepressant use compared to the original analysis, in line with previous research that has shown a higher incidence of depression within the first 6 months of using HC. See Supplementary material 7. A brief discussion on these results has been added in the revised manuscript.

Ideally, we should perform a MAIHDA defining strata according four categories of HC exposure, comparing:(i) new users, (ii) continuous users, (iii) past users and (iv) never-users.. However, further stratification would create many empty strata and unreliable results. Balancing parsimonious and readily interpretable results with appropriate fine-graining of the data is

always a challenge, but we argue that a subclassification to this extent would cloud rather than clarify our results.

Another possible approach to properly answer the referee would be to perform a survival analysis studying individual use of antidepressant along time in non-user and in new-user of HC. However, the information on medication use is rather coarse and the implementation of survival analysis with MAIHDA would considerably complicate the methodology and interpretation of the results. This reasoning has been added to the limitations section in the revised manuscript.

In summary, our study contributes to the research field but by combining new users, continuous users, and past users in the category of “users” our results may underestimate the risk of antidepressant use in new users and overestimate it in continuous users. We have added this information in the discussion part of the revised paper.

Reviewer: 3

Prof. Øjvind Lidegaard, Rigshospitalet, University of Copenhagen, DK-2100 Copenhagen, Denmark

Comments to the Author:

I think the authors have addressed and clarified the concerns expressed in my primary review of the paper. The issue is important, and the study provides new insight into the complexity of factors influencing use of antidepressants with use of hormonal contraception.

Appropriate sensitivity analyses were made to address the issue of length of use, and the design is now clearly explained as being a historical prospective cohort study.

Can be published as it is.

We are glad you now find the paper ready for publication. We made minor further changes in response to another reviewer’s final comments, see above. Thank you for your help in improving our paper.

VERSION 3 – REVIEW

REVIEWER	Roberts, Timothy University of Missouri, Pediatrics
REVIEW RETURNED	08-Sep-2021
GENERAL COMMENTS	All concerns have been addressed. I feel this paper is ready for publication. Excellent work.